# DARNet: Dual Attention Refinement Network with Spatiotemporal Construction for Auditory Attention Detection

**Sheng Yan**[1*]   **Cunhang Fan**[1*†]   **Hongyu Zhang**[1]   **Xiaoke Yang**[1]   **Jianhua Tao**[2]

**Zhao Lv**[1]

[1]Anhui Province Key Laboratory of Multimodal Cognitive Computations,
School of Computer Science and Technology, Anhui University
[2]Department of Automation, Tsinghua University

## Abstract

At a cocktail party, humans exhibit an impressive ability to direct their attention. The auditory attention detection (AAD) approach seeks to identify the attended speaker by analyzing brain signals, such as EEG signals. However, current AAD algorithms overlook the spatial distribution information within EEG signals and lack the ability to capture long-range latent dependencies, limiting the model's ability to decode brain activity. To address these issues, this paper proposes a dual attention refinement network with spatiotemporal construction for AAD, named DARNet, which consists of the spatiotemporal construction module, dual attention refinement module, and feature fusion & classifier module. Specifically, the spatiotemporal construction module aims to construct more expressive spatiotemporal feature representations, by capturing the spatial distribution characteristics of EEG signals. The dual attention refinement module aims to extract different levels of temporal patterns in EEG signals and enhance the model's ability to capture long-range latent dependencies. The feature fusion & classifier module aims to aggregate temporal patterns and dependencies from different levels and obtain the final classification results. The experimental results indicate that DARNet achieved excellent classification performance, particularly under short decision windows. While maintaining excellent classification performance, DARNet significantly reduces the number of required parameters. Compared to the state-of-the-art models, DARNet reduces the parameter count by 91%. Code is available at: https://github.com/fchest/DARNet.git.

## 1   Introduction

The auditory attention detection (AAD) aims to study human auditory attention tendencies by analyzing brain signals [1, 2, 3]. The auditory attention refers to the ability of individuals to isolate or concentrate on specific sounds, which aids them in focusing on a single speaker amidst a multi-speaker environment, a scenario commonly referred to as the "cocktail party scenario" [4]. However, this ability may diminish or even completely disappear for individuals with impairment. Therefore, finding solutions to assist these individuals in overcoming this challenge has become an urgent matter.

Mesgarani and Chang [5] have demonstrated a close connection between auditory attention and brain activity, which indicates that researchers can study auditory attention by analyzing brain activity.

---

[*]Equal contribution.
[†]Corresponding author. Correspondence to cunhang.fan@ahu.edu.cn.

38th Conference on Neural Information Processing Systems (NeurIPS 2024).

Following this concept, many methods such as electrocorticography (ECoG) [5], magnetoencephalography [6, 7], and electroencephalography (EEG) [8, 9] have been used to implement auditory attention detection. Among these methods, EEG-based approaches are widely applied in AAD due to their high temporal resolution, non-invasive mode, and excellent maneuverability [9, 10, 11].

According to the conclusions of Mesgarani and Chang [5], previous studies have utilized stimulus-reconstruction or speech envelope reconstruction methods, which necessitate clean auditory stimuli as input [12, 13]. However, in most real-world scenarios, environments consist of multiple sounds simultaneously. Listeners are exposed to a mixture of these sounds, posing a challenge in obtaining clean auditory stimuli. Therefore, in recent years, the academic community has increasingly focused solely on utilizing EEG signals as input for AAD research [14, 15, 16]. The research method proposed in this paper also exclusively utilizes EEG signals.

Traditional AAD tasks relied on linear models to process EEG signals [17, 18]. However, brain activity is inherently nonlinear, posing challenges for linear models in capturing this complexity. Consequently, they necessitate longer decision windows to extract brain activity features [19]. Some previous studies have indicated that decent decoding performance can be achieved by analyzing different spatial distribution features within each frequency band. These studies project the extracted differential entropy (DE) values onto 2D topological maps and decode them with convolutional neural networks [20, 15]. However, EEG signals are fundamentally time-series data, these methods overlook the dynamic temporal patterns of EEG signals. Other studies analyze EEG signals only in the time domain. For instance, they use long short-term memory (LSTM) networks to capture dependencies within EEG signals and achieve decent decoding performance [2]. However, these studies only focus on the temporal information within EEG signals, neglecting the spatial distribution features, which reflect the dynamic patterns of different brain regions when receiving, processing, and responding to auditory stimuli. Meanwhile, numerous noise points and outliers make it difficult to capture long-range latent dependencies.

To address these issues, this paper proposes a dual attention refinement network with spatiotemporal construction for AAD, named DARNet, which effectively captures the spatiotemporal features and long-range latent dependencies of EEG signals. Specifically, our model consists of three modules: (1) *Spatiotemporal Construction Module*. The spatiotemporal construction module employs a temporal convolutional layer and a spatial convolutional layer. The temporal convolutional layer effectively captures the temporal dynamic features of EEG signals, and the spatial convolutional layer captures the spatial distribution features among different channels, thereby constructing a robust embedding for the next layer. (2) *Dual Attention Refinement Module*. The dual-layer self-attention refinement module consists of two layers, each comprising a multi-head self-attention and a refinement layer. This design is intended to capture long-range latent dependencies and deeper sequence patterns in EEG signals. (3) *Feature Fusion & Classifier Module*. The attention features generated by the dual-layer self-attention refinement module, comprising both shallow and deep levels, are fed into the feature fusion module to obtain richer representations, enhancing the model's robustness and generalization. The fused features are input into a classifier to predict the auditory attention tendencies of the subjects.

To this end, We evaluated the decoding performance of DARNet on three datasets: DTU, KUL, and MM-AAD. The results demonstrate that DARNet outperforms the current state-of-the-art model on all three datasets. The main contributions of this paper are summarized as follows:

- We propose a novel auditory attention decoding architecture, which consists of a spatiotemporal construction module, a dual attention refinement module, and a feature fusion module. This architecture could fully leverage the spatiotemporal features and capture long-range latent dependencies of EEG signals.
- The DARNet achieves remarkable decoding accuracy within very short decision windows, surpassing the current state-of-the-art (SOTA) model by 5.9% on the DTU dataset and 4.9% on the KUL dataset, all under a 0.1-second decision window. Furthermore, compared to the current state-of-the-art model with 0.91 million training parameters, DARNet achieves further parameter reduction, requiring only 0.08 million parameters.

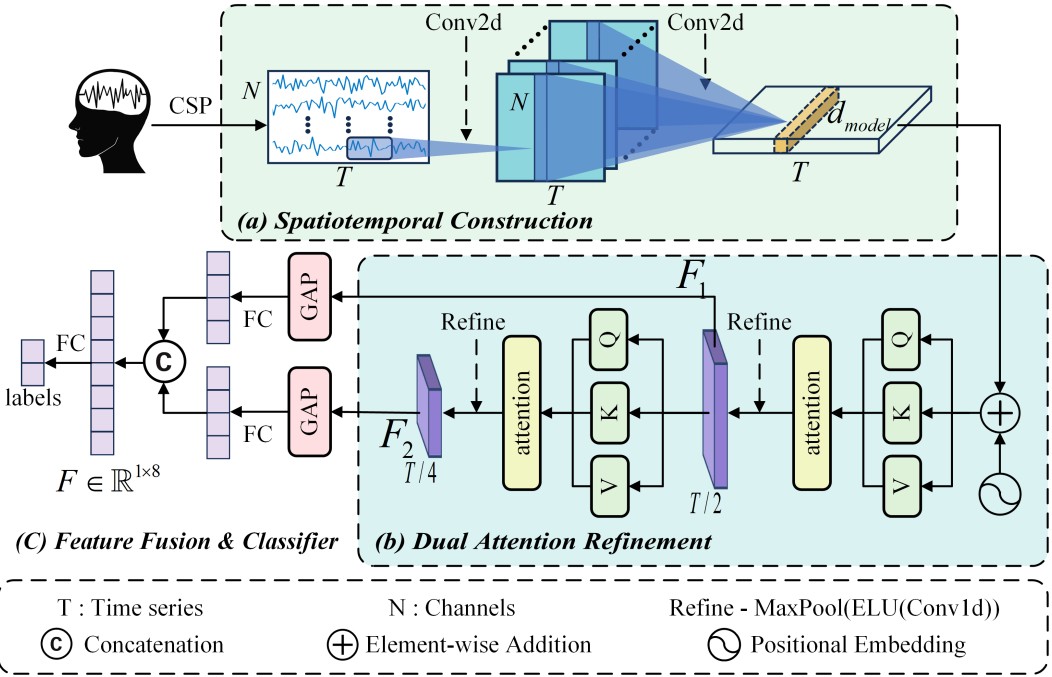

Figure 1: The framework of the DARNet model for AAD, which mainly consists of three modules: (a) spatiotemporal construction module, (b) dual attention refinement module, and (c) feature fusion & classifier module. The model inputs are common spatial patterns (CSP) extracted from EEG signals, and the outputs are two predicted labels related to auditory attention.

## 2 Methodology

The previous AAD methods overlooked the influence of spatial distribution characteristics on decoding performance and struggled to capture the long-range dependencies in EEG signals [14, 20]. To address these issues, we proposed DARNet, which consists of a spatiotemporal construction module, a dual attention refinement module, and a feature fusion & classifier module, see Figure 1. Our proposed DARNet effectively captures the spatiotemporal features of EEG signals and has the capability to capture long-range latent dependencies in EEG signals.

By employing a moving window on the EEG data, we obtain a series of decision windows, each containing a small duration of EEG signals. Let $R = [r_1, ..., r_i, ..., r_N] \in \mathbb{R}^{T \times N}$ represents the EEG signals of a decision window, where $r_i \in \mathbb{R}^{N \times 1}$ represents the EEG data at the $i$-th time point within a decision window, contains $N$ channels. Here $N$ represents the number of EEG channels and $T$ denotes the length of the decision window. Before inputting EEG data into the DARNet, we employ a common spatial patterns (CSP) algorithm to extract raw features from the EEG data under different brain states [21, 22].

$$E = CSP(R) \in \mathbb{R}^{c\_in \times T} \tag{1}$$

where $CSP(\cdot)$ represents the CSP algorithm, $E \in \mathbb{R}^{N \times T}$ represents the processed EEG signal. $c\_in$ is the components of the CSP algorithm and T denotes the length of the decision window.

### 2.1 Spatiotemporal Construction Module

EEG signals record the brain's neuronal electrical activity, varying over time and reflecting activity patterns and connectivity across brain regions [23]. By constructing spatiotemporal features from EEG signals, it's possible to analyze the brain's response patterns to auditory stimuli. However, previous studies only focused on local temporal patterns in EEG data, overlooking the spatial distribution features. Therefore, in addition to the conventional use of temporal filters, we introduced a spatial filter [24] to construct the spatiotemporal features of EEG signals.

Firstly, we use temporal convolution layers to capture the instantaneous changes in EEG signals, thereby constructing the temporal patterns $E_t$ of the EEG signals. It can be formulated as follows:

$$E_t = GELU(TemporalConv2d(E)) \in \mathbb{R}^{4d_{model} \times c\_in \times T} \tag{2}$$

where $TemporalConv2d(\cdot)$ performs an 2-D convolutional filters (kernel size=$1 \times 8$) on time dimension with $GELU(\cdot)$ activation function. $d_{model}$ represents the embedding dimension.

Subsequently, we employ a spatial convolutional layer with a receptive field spanning all channels to capture the spatial distribution features $S$ of EEG signals across different channels, thereby aiding the model in comprehensively understanding the brain's activity patterns in response to various auditory stimuli.

$$S = GELU(SpatialConv2d(E_t)) \in \mathbb{R}^{d_{model} \times T} \tag{3}$$

where $SpatialConv2d(\cdot)$ performs an 2-D convolutional filters with a $c\_in \times 1$ kernel size on spatial dimension. By doing so, we not only capture the temporal patterns in EEG signals but also integrate the spatial distribution characteristics of EEG signals, thereby constructing input embedding $S$ containing comprehensive spatiotemporal information for the next layer. This integrated input better reflects the complex features within EEG signals, providing richer information for subsequent analysis and processing.

## 2.2 Dual Attention Refinement Module

Previous psycho-acoustic research has demonstrated that human attention is a dynamic and time-related activity [25, 26]. The brain activity from the preceding moment can profoundly influence subsequent brain activity [27]. However, previous AAD algorithms were hindered by model depth and the noise and outliers in EEG data, making them ineffective at capturing the long-range latent dependencies in EEG signals.

To address this issue, we proposed a dual self-attention mechanism, which has greater potential for capturing long-range latent dependencies and deeper sequence patterns in EEG signals. Inspired by Zhou et al. [28], Yu et al. [29], We introduced a self-attention refinement operation, which refines the dominant temporal features through convolution and pooling operations, compressing the original EEG series of length $T$ to half its length. This self-attention refinement operation reduces the impact of noise and outliers, while also decreasing the model's parameter count. This enhances the model's generalization and robustness. The single-layer attention refinement module can be formulated as follows:

$$F = MaxPool(ELU(Conv1d(MultiHeadAttention(x)))) \tag{4}$$

where $MultiHeadAttention(\cdot)$ denotes multi-head self-attention algorithm [30], $Conv1d(\cdot)$ represents an 1-D convolutional filters (kernel width=3) on time dimension. The $ELU(\cdot)$ is the activation function proposed by Clevert et al. [31], $MaxPool(\cdot)$ denotes a max-pooling layer with stride 2.

Before applying the temporal attention feature extraction module, we add the absolute positional embedding [30] to the input embedding $S$ as follows:

$$s_i = s_i + p_i \tag{5}$$

where $s_i$ represents the embedding vector of the $i_{th}$ time step, $p_i \in \mathbb{R}^{d_{model}}$ represents $i_{th}$ time step position.

To obtain different levels of temporal features from EEG signals and to capture the long-range latent dependencies, we stacked two of the above attention refinement extraction modules.

$$F_1 = MaxPool(ELU(Conv1d(MultiHeadAttention(S)))) \in \mathbb{R}^{d_{model} \times \frac{T}{2}} \tag{6}$$

$$F_2 = MaxPool(ELU(Conv1d(MultiHeadAttention(F_1)))) \in \mathbb{R}^{d_{model} \times \frac{T}{4}} \tag{7}$$

where $F_1$ and $F_2$ contain different levels of dependencies and temporal patterns in the EEG signals, respectively.

## 2.3 Feature Fusion & Classifier Module

Features at different levels can reflect various characteristics of the pattern. By optimizing and combining these different features, it preserves effective discriminative information from features

at different levels while also to some extent eliminating redundant information [32]. Therefore, we designed a feature fusion module as follows:

First, we project the features $F_1$ and $F_2$ into the same dimension.

$$F_1' = Linear(AdaptiveAvgPool(F_1)) \in \mathbb{R}^4 \tag{8}$$

$$F_2' = Linear(AdaptiveAvgPool(F_2)) \in \mathbb{R}^4 \tag{9}$$

where $AdaptiveAvgPool(\cdot)$ denotes an adaptive average pooling layer, $Linear$ denotes a linear layer.

Second, We concatenate feature $F_1'$ and $F_2'$ to obtain the fused feature vector $F$.

$$F = [F_1', F_2'] \tag{10}$$

Finally, we employ a fully connected layer to obtain the final auditory attention prediction.

$$p = w(F + b) \tag{11}$$

where $w$ and $b$ are the weight and the bias of the fully connected layer, $p$ denotes the predicted direction label. In the training stage, we employ the cross entropy loss function to supervise the network training.

## 3 Experiments

### 3.1 Dataset

In this section, we conduct experiments on three publicly available datasets, namely KUL [33, 34], DTU [35, 36] and MM-AAD [20], which are commonly used in auditory attention detection to evaluate the effectiveness of our DARNet. KUL and DTU only contain EEG data of the auditory stimulus scenes. MM-AAD contains EEG data of the audio-only scene and the audio-visual scene. We summarize the details of the above datasets in Table 1.

1) **KUL Dataset:** In this dataset, 64-channel EEG data were collected from 16 normal-hearing subjects using a BioSemi ActiveTwo device at a sampling rate of 8,192 Hz in a soundproof room. Each subject was instructed to focus on one of two simultaneous speakers. The auditory stimuli were filtered at 4kHz and set at 60dB through in-ear headphones, which contain four Dutch short stories, narrated by three male Flemish speakers. Two listening conditions were employed: dichotic (dry) presentation with one speaker per ear, and head-related transfer function (HRTF) filtered presentation, simulating speech from 90° left or right. Each subject listened to 8 trials, which lasted 6 minutes.

2) **DTU Dataset:** In this dataset, 64-channel EEG data were collected from 18 normal-hearing subjects using a BioSemi ActiveTwo device at a sampling rate of 512 Hz. Each subject was instructed to focus on one of two simultaneous speakers, who presented at 60° relative to the subject. The auditory stimuli were set at 60dB through ER-2 earphones, which contain Danish audiobooks, narrated by three male speakers and three female speakers. Each subject listened to 60 trials, which lasted 50 seconds.

3) **MM-AAD Dataset:** In this dataset, 32-channel EEG data were collected from 50 normal-hearing subjects (34 males and 16 females) at a sampling rate of 4kHz, following the 10/20 international system. Each subject was exposed to both audio-only and audio-visual stimuli. They were instructed to focus on one of two simultaneous speakers, who presented at left or right spatial direction relative to the subject. The auditory stimuli comprised 40 classic Chinese stories narrated by both male and female speakers. Each subject listened to 20 trials, which lasted 165 seconds.

### 3.2 Data Processing

To fairly compare the performance of the proposed DARNet model, specific preprocessing steps are applied to each dataset (KUL, DTU, and MM-AAD). For the KUL dataset, the EEG data were firstly re-referenced to the average response of mastoid electrodes, then bandpass filtered between 0.1 Hz

Table 1: Details of three datasets used in the experiments.

| Dataset | Subjects | Scene | Language | Duration per subject (minutes) | Total duration (hours) |
|---|---|---|---|---|---|
| KUL | 16 | audio-only | Dutch | 48 | 12.8 |
| DTU | 18 | audio-only | Danish | 50 | 15.0 |
| MM-AAD | 50 | audio-only | Chinese | 55 | 45.8 |
| | 50 | audio-visual | Chinese | 55 | 45.8 |

and 50 Hz, and finally down-sampled to 128 Hz. For the DTU dataset, the EEG data were filtered to remove 50 Hz linear noise and harmonics. Eye artifacts were eliminated through joint decorrelation and the EEG data were re-referenced to the average response of mastoid electrodes. Finally, the EEG data were down-sampled to 64 Hz. For the MM-AAD dataset, the EEG data were firstly bandpass filtered between 0.1 Hz and 50 Hz, then removed 50 Hz noise through a notch filter. Additionally, eye artifacts were eliminated, and further noise removal was achieved, using independent component analysis (ICA). Finally, the EEG data were down-sampled to 128 Hz.

We evaluated our proposed DARNet model and compared it with other state-of-the-art models under three decision window lengths: 0.1s, 1s, and 2s. Specifically, we selected three publicly available models as our baseline for comparison: SSF-CNN [37], MBSSFCC [15], and DBPNet [20].

## 3.3 Implement Details

In previous AAD research, the accuracy of auditory attention prediction classification has been used as a benchmark for model performance. We followed this convention and evaluated our proposed DARNet on the KUL, DTU, and MM-AAD datasets. As follows, we take the KUL dataset with a 1-second decision window as an example to illustrate implementation details, including training settings and network configuration.

Firstly, we set the proportions of the training, validation, and test sets to 8:1:1. For each subject of the KUL dataset, we get 4,600 decision windows for training, 576 decision windows for validation, and 576 decision windows for testing. Meanwhile, we set the batch size to 32, the maximum number of epochs to 100, and employ an early stopping strategy. Training will stop if the loss function value on the validation set does not decrease for 10 consecutive epochs. Additionally, we utilize the Adam optimizer with a learning rate of 5e-4 and weight decay of 3e-4 to train the model. The DARNet is performed using PyTorch.

Before inputting EEG data into the DARNet, we employ the CSP algorithm to extract raw features $E \in \mathbb{R}^{128 \times 64}$ from the EEG data. The data is transposed and expanded, represented as $E' \in \mathbb{R}^{1 \times 64 \times 128}$. Then, through the spatiotemporal construction module ($c_{in}$ is set to 16), we can get embedding data $S \in \mathbb{R}^{16 \times 1 \times 128}$. After dimensionality reduction, transposition, and the addition of absolute positional embedding, the data is fed into the dual attention refinement module, resulting in two distinct level features, $F_1 \in \mathbb{R}^{16 \times 64}$ and $F_2 \in \mathbb{R}^{16 \times 32}$. The $F_1$ and $F_2$ are sent to the feature fusion module, where they undergo global average pooling and dimensionality reduction via a fully connected (FC) layer (input: 16, output: 4) before being concatenated to obtain the fused feature, $F \in \mathbb{R}^8$. Finally, $F$ is passed through another FC layer (input: 8, output:2) to obtain the final auditory attention prediction $p \in \mathbb{R}^2$.

## 4 Result

### 4.1 Performance of DARNet

To evaluate the performance of DARNet, we conducted comprehensive experiments under decision windows of 0.1-second, 1-second, and 2-second, respectively, as shown in Figure 2. Additionally, We compared our DARNet with other advanced models, as shown in Table 2. The results are replicated from the corresponding papers.

DARNet has outperformed the current state-of-the-art models on the KUL, DTU, and MM-AAD datasets, achieving further enhancements in performance. On the KUL dataset, the DARNet achieves

Table 2: Auditory attention detection accuracy(%) comparison on DTU, KUL and MM-AAD dataset. The results annotated by * are taken from [20]. Our experimental setup is consistent with theirs to ensure fairness in comparison. Hence, we directly cited their results.

| Dataset | Scene | Model | Decision Window | | |
|---|---|---|---|---|---|
| | | | 0.1-second | 1-second | 2-second |
| KUL | audio-only | SSF-CNN* [37] | $76.3 \pm 8.47$ | $84.4 \pm 8.67$ | $87.8 \pm 7.87$ |
| | | MBSSFCC* [15] | $79.0 \pm 7.34$ | $86.5 \pm 7.16$ | $89.5 \pm 6.74$ |
| | | BSAnet [38] | - | $93.7 \pm 4.02$ | $95.2 \pm 3.08$ |
| | | DenseNet-3D [39] | - | $94.3 \pm 4.3$ | $95.9 \pm 4.3$ |
| | | DBPNet* [20] | $87.1 \pm 6.55$ | $95.0 \pm 4.16$ | $96.5 \pm 3.50$ |
| | | **DARNet(ours)** | $\mathbf{91.6 \pm 4.83}$ | $\mathbf{96.2 \pm 3.04}$ | $\mathbf{97.2 \pm 2.50}$ |
| DTU | audio-only | SSF-CNN* [37] | $62.5 \pm 3.40$ | $69.8 \pm 5.12$ | $73.3 \pm 6.21$ |
| | | MBSSFCC* [15] | $66.9 \pm 5.00$ | $75.6 \pm 6.55$ | $78.7 \pm 6.75$ |
| | | BSAnet [38] | - | $83.1 \pm 6.75$ | $85.6 \pm 6.47$ |
| | | EEG-Graph Net [40] | $72.5 \pm 7.41$ | $78.7 \pm 6.47$ | $79.4 \pm 7.16$ |
| | | DBPNet* [20] | $75.1 \pm 4.87$ | $83.9 \pm 5.95$ | $86.5 \pm 5.34$ |
| | | **DARNet(ours)** | $\mathbf{79.5 \pm 5.84}$ | $\mathbf{87.8 \pm 6.02}$ | $\mathbf{89.9 \pm 5.03}$ |
| MM-AAD | audio-only | SSF-CNN* [37] | $56.5 \pm 5.71$ | $57.0 \pm 6.55$ | $57.9 \pm 7.47$ |
| | | MBSSFCC* [15] | $75.3 \pm 9.27$ | $76.5 \pm 9.90$ | $77.0 \pm 9.92$ |
| | | DBPNet* [20] | $91.4 \pm 4.63$ | $92.0 \pm 5.42$ | $92.5 \pm 4.59$ |
| | | **DARNet(ours)** | $\mathbf{94.9 \pm 4.79}$ | $\mathbf{96.0 \pm 4.00}$ | $\mathbf{96.5 \pm 3.59}$ |
| | audio-visual | SSF-CNN* [37] | $56.6 \pm 3.82$ | $57.2 \pm 5.59$ | $58.2 \pm 6.39$ |
| | | MBSSFCC* [15] | $77.2 \pm 9.01$ | $78.1 \pm 10.1$ | $78.4 \pm 9.57$ |
| | | DBPNet* [20] | $92.1 \pm 4.47$ | $92.8 \pm 5.94$ | $93.4 \pm 4.86$ |
| | | **DARNet(ours)** | $\mathbf{95.8 \pm 4.04}$ | $\mathbf{96.4 \pm 3.72}$ | $\mathbf{96.8 \pm 3.44}$ |

average accuracies of 91.6% (SD: 4.83%), 96.2% (SD: 3.04%), 97.2% (SD: 2.50%) under 0.1-second, 1-second and 2-second decision window, respectively. On the DTU dataset, the DARNet achieves average accuracies of 79.5% (SD: 5.84%) for 0.1-second decision window, 87.8% (SD: 6.02%) for 1-second decision window, 89.9% (SD: 5.03%) for 2-second decision window, respectively. On the MM-AAD dataset, the DARNet also demonstrates outstanding decoding accuracies of 94.9% (SD: 4.79%) for 0.1-second, 96.0% (SD: 4.00%) for 1-second, 96.5% (SD: 3.59%) for 2-second in the audio-only scene, and 95.8% (SD: 4.04%) for 0.1-second, 96.4% (SD: 3.72%) for 1-second, 96.8% (SD: 3.44%) for 2-second in the audio-visual scene.

Overall, DARNet's decoding accuracy increases with larger decision windows, consistent with prior research [15, 14]. This is because longer decision windows provide more information for the model to make judgments while also mitigating the impact of individual outliers on the predictions. However, DARNet still maintains excellent performance under the 0.1-second decision window. Additionally, we observe that in the MM-AAD dataset, performance is better in the audio-visual condition compared to the audio-only condition in two different scenarios. We attribute this improvement to the visual cues aiding humans in localizing sound sources.

## 4.2 Ablation Study

We conducted comprehensive ablation experiments by removing the spatial feature extraction module, the temporal feature extraction module, and the feature fusion module. Additionally, we supplemented our study with ablation experiments using a single-layer attention refinement module on the KUL and DTU dataset, referred to as single-DARNet. All experimental conditions remained the same as in previous settings. Additionally, we ensured that all model network parameters were fully optimized to guarantee that the model's performance reached its best under each condition, whether a module was removed or added. The results of the ablation experiments are shown in Table 3.

Experimental results show that on the DTU dataset, after removing the spatial feature extraction module from DARNet, the average accuracy decreased by 10.1% under a 0.1s decision window,

Table 3: Ablation Study on KUL, DTU, and MM-AAD dataset.

| Dataset | Scene | Model | Decision Window | | |
|---|---|---|---|---|---|
| | | | 0.1-second | 1-second | 2-second |
| KUL | audio-only | w/o spatial feature | 81.1 ± 6.51 | 87.5 ± 7.24 | 89.0 ± 4.93 |
| | | w/o temporal feature | 81.2 ± 6.94 | 86.7 ± 7.30 | 89.1 ± 5.37 |
| | | w/o feature fusion | 90.5 ± 5.09 | 95.2 ± 3.58 | 96.1 ± 3.46 |
| | | single-DARNet | 91.1 ± 5.18 | 95.5 ± 3.28 | 96.2 ± 2.98 |
| | | **DARNet(ours)** | **91.6 ± 4.83** | **96.2 ± 3.04** | **97.2 ± 2.50** |
| DTU | audio-only | w/o spatial feature | 71.5 ± 5.79 | 76.3 ± 7.21 | 79.1 ± 6.79 |
| | | w/o temporal feature | 70.9 ± 5.62 | 75.3 ± 6.62 | 79.2 ± 6.84 |
| | | w/o feature fusion | 77.5 ± 7.07 | 86.3 ± 6.16 | 88.6 ± 5.71 |
| | | single-DARNet | 79.1 ± 5.66 | 86.3 ± 5.83 | 88.0 ± 4.03 |
| | | **DARNet(ours)** | **79.5 ± 5.84** | **87.8 ± 6.02** | **89.9 ± 5.03** |
| MM-AAD | audio-only | w/o spatial feature | 90.0 ± 5.76 | 91.0 ± 5.38 | 92.5 ± 4.76 |
| | | w/o temporal feature | 87.8 ± 5.66 | 89.0 ± 5.20 | 91.1 ± 5.31 |
| | | w/o feature fusion | 94.3 ± 3.94 | 94.8 ± 4.05 | 95.7 ± 4.21 |
| | | **DARNet(ours)** | **94.9 ± 4.79** | **96.0 ± 4.00** | **96.5 ± 3.59** |
| | audio-visual | w/o spatial feature | 90.4 ± 5.99 | 91.2 ± 5.56 | 93.1 ± 5.36 |
| | | w/o temporal feature | 89.4 ± 6.96 | 90.5 ± 6.04 | 92.1 ± 5.72 |
| | | w/o feature fusion | 95.3 ± 4.57 | 95.7 ± 3.88 | 96.1 ± 3.96 |
| | | **DARNet(ours)** | **95.8 ± 4.04** | **96.4 ± 3.72** | **96.8 ± 3.44** |

13.1% under a 1s decision window, and 12.0% under a 2s decision window. After removing the temporal feature extraction module, the average accuracy for the 0.1s, 1s, and 2s decision windows decreased by 10.9%, 14.2%, and 11.9%, respectively. After removing the feature fusion module, the average accuracy decreased by 2.5% under a 0.1s decision window, 1.6% under a 1s decision window, and 1.4% under a 2s decision window. On the KUL dataset and the MM-AAD dataset, removing the aforementioned modules also resulted in similar trends of decreased average accuracy.

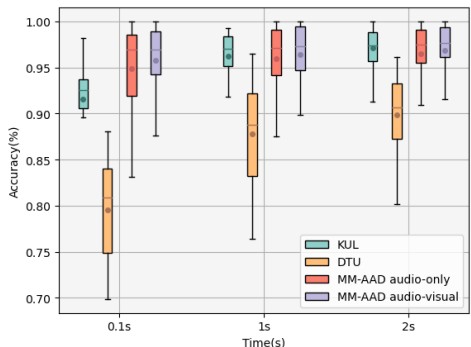

Figure 2: AAD accuracy(%) of DARNet across all subjects on three datasets.

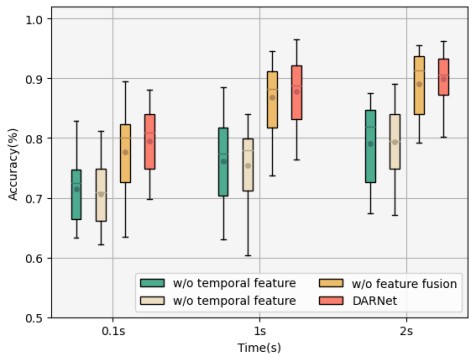

Figure 3: AAD accuracy(%) of the ablation study across all subjects on the DTU dataset.

## 4.3 Experimental Correction

To ensure fairness in comparisons, we aligned our previous experimental setup and data processing methods with those of DBPNet. However, during the code review for the final version of this paper, we found that DBPNet applies CSP to the data prior to dataset splitting, which can lead to data leakage. Consequently, we corrected the processing steps and conducted additional experiments, as shown in Table 4. The data preprocessing steps for the supplementary experiments are as follows:

1. Split each trial of the dataset into the first 90% for training and the last 10% for testing.

2. Fit the CSP transformation matrix using the training data, project the data to extract features, and apply the matrix to the testing data.

3. Apply a sliding window with 50% overlap to the processed training data, then randomly split into a training (90%) and a validation (10%) set. Similarly, apply a sliding window to the processed testing data as the final test set.

Following these steps, we rigorously avoided risk of data leakage. Experimental results show that both corrected DBPNet[2] and DARNet[2] exhibit a performance decline compared to the original DBPNet[1] and DARNet[1]. However, DARNet[2] still achieves state-of-the-art performance under most test conditions, as shown in Table 4. For instance, on the KUL dataset with a 0.1-second decision window, DARNet[2] maintains a decoding accuracy of 89.2%.

Table 4: Experimental Correction on KUL, DTU, and MM-AAD dataset. The experimental setup for the results marked with [1] is consistent with DBPNet[20], where CSP is applied prior to dataset splitting. In contrast, for the results marked with [2], CSP is applied after dataset splitting.

| Dataset | Scene | Model | Decision Window | | |
|---------|-------|-------|-----------------|---|---|
| | | | 0.1-second | 1-second | 2-second |
| KUL | audio-only | DBPNet[1] | $87.1 \pm 6.55$ | $95.0 \pm 4.16$ | $96.5 \pm 3.50$ |
| | | DBPNet[2] | $85.3 \pm 6.22$ | $94.4 \pm 4.62$ | $95.3 \pm 4.63$ |
| | | DARNet[1] | $91.6 \pm 4.83$ | $96.2 \pm 3.04$ | $97.2 \pm 2.50$ |
| | | DARNet[2] | $89.2 \pm 5.50$ | $94.8 \pm 4.53$ | $95.5 \pm 4.89$ |
| DTU | audio-only | DBPNet[1] | $75.1 \pm 4.87$ | $83.9 \pm 5.95$ | $86.5 \pm 5.34$ |
| | | DBPNet[2] | $74.0 \pm 5.20$ | $79.8 \pm 6.91$ | $80.2 \pm 6.79$ |
| | | DARNet[1] | $79.5 \pm 5.84$ | $87.8 \pm 6.02$ | $89.9 \pm 5.03$ |
| | | DARNet[2] | $74.6 \pm 6.09$ | $80.1 \pm 6.85$ | $81.2 \pm 6.34$ |
| MM-AAD | audio-only | DBPNet[1] | $91.4 \pm 4.63$ | $92.0 \pm 5.42$ | $92.5 \pm 4.59$ |
| | | DBPNet[2] | $90.0 \pm 5.51$ | $90.7 \pm 5.68$ | $91.6 \pm 4.82$ |
| | | DARNet[1] | $94.9 \pm 4.79$ | $96.0 \pm 4.00$ | $96.5 \pm 3.59$ |
| | | DARNet[2] | $91.5 \pm 5.27$ | $92.2 \pm 4.54$ | $92.8 \pm 5.22$ |
| | audio-visual | DBPNet[1] | $92.1 \pm 4.47$ | $92.8 \pm 5.94$ | $93.4 \pm 4.86$ |
| | | DBPNet[2] | $92.0 \pm 5.51$ | $93.0 \pm 5.19$ | $92.7 \pm 6.04$ |
| | | DARNet[1] | $95.8 \pm 4.04$ | $96.4 \pm 3.72$ | $96.8 \pm 3.44$ |
| | | DARNet[2] | $92.7 \pm 5.34$ | $93.4 \pm 5.23$ | $94.2 \pm 4.84$ |

# 5 Discussion

## 5.1 Comparative Analysis

To further evaluate the performance of our proposed DARNet, we compared it with other advanced AAD models, as shown in Table 2. The results indicate that our DARNet has achieved a significant improvement over the current state-of-the-art results.

For example, on the DTU dataset, our DARNet has shown relative improvements of 27.2%, 18.8%, 9.7%, and 5.9% for 0.1-second decision window, compared to the SSF-CNN, MBSSFCC, EEG-Graph Net and DBPNet models, respectively. Compared to the SSF-CNN, MBSSFCC, BSAnet, EEG-Graph Net, and DBPNet models, the relative improvements achieve 25.8%, 16.1%, 5.7%, 11.6%, 4.6% for 1-second decision window, and 22.6%, 14.2%, 5.0%, 13.2%, 3.9% for 2-second. On both the KUL and MM-AAD datasets, DARNet has achieved similar improvements compared to the state-of-the-art models. The particularly outstanding results achieved across all three datasets under the 0.1-second decision window indicate the potential of DARNet for real-time decoding of auditory attention.

Overall, the excellent performance of DARNet across different datasets and decision windows demonstrates its robustness and versatility in various contexts. This further validates the potential of DARNet as an effective EEG analysis model and provides strong support for its widespread application in real-world scenarios.

## 5.2 Ablation Analysis

As shown in Table 3 and Figure 3, compared with removing the spatial feature extraction step, removing the temporal feature extraction step, removing the feature fusion module, and using a single-layer attention refinement module, we believe DARNet performs excellently for the following reasons:

**1. Integrating multiple sources of information:** DARNet integrates temporal and spatial distribution features from EEG signals, constructing richer and more robust spatiotemporal features. This enables the model to comprehensively understand the spatiotemporal information within EEG signals, thereby enhancing the understanding of brain activity. In contrast, removing any single feature may lead to information loss or the inability to capture the transient changes in EEG signals, thereby impacting the model's performance.

**2. Comprehensive capture of temporal dependencies:** The dual attention refinement module and feature fusion module of DARNet comprehensively capture temporal patterns and dependencies at different levels, enabling the model to better understand the temporal dynamics within EEG signals. This holistic consideration of features at different time scales is crucial for the analysis of EEG data.

**3. Robust feature representation:** Despite observing that removing the feature fusion module did not lead to a significant decrease in accuracy across the three datasets, the performance variability of DARNet increases substantially. We believe that the feature fusion module integrates temporal patterns and dependencies at different levels, enabling the model to better understand and utilize the complex relationships within the data, thus enhancing the robustness and generalization of the model.

Table 5: The training parameter counts comparison. "M" denotes a million.

| Model | Trainable Parameters Counts |
|---|---|
| SSF-CNN [37] | 4.21M |
| MBSSFCC [15] | 83.91M |
| DBPNet [20] | 0.91M |
| **DARNet (ours)** | **0.08M** |

## 5.3 Computational Cost

We compare the training parameter counts of our DARNet, SSF-CNN [37], MBSSFCC [15], and DBPNet [20], with the results shown in Table 5. The parameter count of DARNet is 51.6 times lower than that of SSF-CNN, 1331.5 times lower than that of MBSSFCC, and 10.4 times lower than that of DBPNet. Compared to other models, DARNet demonstrates superior parameter efficiency. Despite having fewer parameters, DARNet maintains good performance, indicating its ability to be applied in resource-constrained environments for AAD analysis, thus demonstrating practical utility.

# 6 Conclusion

In this paper, we propose the DARNet, a novel dual attention refinement network with spatiotemporal construction for auditory attention detection. By employing spatial convolution operations across all channels, DARNet effectively leverages the spatial information embedded in EEG signals, thereby constructing a more robust spatiotemporal feature. Additionally, DARNet integrates dual attention refinement and feature fusion techniques to comprehensively capture temporal patterns and dependencies at various levels, enhancing the model's ability to capture the temporal dynamics within EEG signals. We evaluate the performance of DARNet on three datasets: KUL, DTU, and MM-AAD. DARNet achieves a decoding accuracy of 96.2% on the 1-second decision window of the KUL dataset and 87.8% on the 1-second decision window of the DTU dataset, demonstrating significant improvements compared to current state-of-the-art models. The experimental results validate the effectiveness and efficiency of the DARNet architecture, indicating its potential for practical applications. In future research, we plan to further explore DARNet's performance on cross-subject tasks to verify its generalization and robustness.

## Acknowledgments and Disclosure of Funding

This work is supported by the STI 2030—Major Projects (No. 2021ZD0201500), the National Natural Science Foundation of China (NSFC) (No.62201002, 6247077204), Excellent Youth Foundation of Anhui Scientific Committee (No. 2408085Y034), Distinguished Youth Foundation of Anhui Scientific Committee (No. 2208085J05), Special Fund for Key Program of Science and Technology of Anhui Province (No. 202203a07020008), Open Fund of Key Laboratory of Flight Techniques and Flight Safety, CACC (No, FZ2022KF15), Cloud Ginger XR-1.

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
