# OpenReview forum: "DARNet: Dual Attention Refinement Network with Spatiotemporal Construction for Auditory Attention Detection"
_NeurIPS.cc/2024/Conference — NeurIPS 2024 poster_

### Official Review · Reviewer_XzJY · 2024-07-03

**Soundness:** 3
**Presentation:** 3
**Contribution:** 3
**Rating:** 8
**Confidence:** 5

**Summary:**

This paper makes a substantial contribution to the field of auditory attention detection (AAD) by presenting a more accurate and efficient model, the dual attention refinement network with spatiotemporal construction (DARNet). The authors effectively address critical limitations in current AAD algorithms, specifically the lack of spatial distribution information and the inability to capture long-range latent dependencies within EEG signals. The methodological innovations and empirical results suggest that DARNet represents a significant advancement in decoding brain activity related to auditory attention tasks. The experiments show significant improvements over state-of-the-art methods.

**Strengths:**

1. This paper uses a simple yet effective method to address the previous auditory attention detection algorithms' insufficient utilization of the spatiotemporal features of EEG signals.
2. The proposed method is technically sound and the idea seems to be interesting.
3. The experimental setting is valid and extensive,  and the experimental results indicate significant improvements over some comparison methods on several datasets.

**Weaknesses:**

Overall, the writing of this paper is relatively clear and the structure is complete. However, there are still some issues, such as grammatical errors and some sentences that are not clearly articulated. 3）There are many writing problems. The authors should further polish this manuscript:
1）Line 100-102: The expression is not very clear.
2）Line 103: "However previous" -> "However, previous".
3）Line 188: "or" -> "and".
4）Line 292: "increased" -> "increases".

**Questions:**

1. Why did you use a dual attention refinement module instead of three layers or more? The authors should provide more analysis
2. Did you lose the FC layer for the final classification part in Figure 1?

**Limitations:**

The DARNet has only been validated under the subject-dependent condition, and it may not perform as well under subject-independent conditions.

---

> ### Author Rebuttal · Authors · 2024-08-06
>
> # Rebuttal:
>
> Thank you so much for your thoughtful comments and the time to provide constructive feedback!
>
> ### Weaknesses:
>
> 1. **Regarding grammatical errors and unclear sentence expressions in the paper:**
>
>    We are very grateful to the Reviewer for carefully reviewing the paper. We have thoroughly examined and corrected the grammatical errors and unclear sentences to ensure the paper is clearer and more accurate.
>
> 2. **For the specific issues you pointed out, we have made the following revisions:**
>
>    - **Lines 100-102:** We have reorganized the sentence to make it clearer. The revised sentence is: "EEG signals record the brain's neuronal electrical activity, varying over time and reflecting activity patterns and connectivity across brain regions."
>    - **Line 103:** We have changed "However previous" to "However, previous."
>    - **Line 188:** We have changed "or" to "and."
>    - **Line 292:** We have changed "increased" to "increases."
>
> ### Questions:
>
> 1. **Why did you use a dual attention refinement module instead of three layers or more?**
>
>    In the field of auditory attention detection, researchers are more focused on addressing the issue of low decoding accuracy in low-latency scenarios. Specifically, they aim to reflect the model's effectiveness in low-latency conditions by examining decoding accuracy within a 0.1-second decision window. However, EEG sequences within this 0.1-second window typically contain 12 or 6 data points, which is insufficient for our model to use three or more layers of refinement. Additionally, after two refinement operations, the EEG sequence is already compressed to 3 or 1 data points, making further compression meaningless. Therefore, we chose to use only a two-layer attention refinement module.
>
>    Furthermore, we conducted ablation experiments, as shown in Table 1 and Table 2. The results indicate that using a single-layer attention refinement compared to a two-layer attention refinement resulted in only a 0.5% decrease in performance, which aligns with our assumptions.
>
>    Table1: Ablation Study on DTU dataset.
>
>    | Model         | 0.1s                             | 1s                                | 2s                               | 5s                               |
>    | :------------ | :------------------------------- | :-------------------------------- | :------------------------------- | :------------------------------- |
>    | single-DARNet | $79.1 \pm 5.66$                  | $86.3 \pm 5.83$                   | $88.0 \pm 4.03$                  | $90.2 \pm 5.62$                  |
>    | **DARNet**    | $\textbf{79.5}\pm \textbf{5.84}$ | $\textbf{87.8} \pm \textbf{6.02}$ | $\textbf{89.9}\pm \textbf{5.03}$ | $\textbf{93.1}\pm \textbf{4.37}$ |
>
>    Table2: Ablation Study on KUL dataset
>
>    | Model         | 0.1s            | 1s              | 2s              | 5s              |
>    | :------------ | :-------------- | :-------------- | :-------------- | :-------------- |
>    | single-DARNet | $91.1 \pm 5.18$ | $95.5 \pm 3.28$ | $96.2 \pm 2.98$ | $96.7 \pm 4.03$ |
>    | **DARNet**    | $91.6\pm 4.83$  | $96.2 \pm 3.04$ | $97.2\pm 2.50$  | $98.0\pm 3.17$  |
>
> 2. Did you lose the FC layer for the final classification part in Figure 1?
>
>    Thank you for your thorough review and feedback. We acknowledge that we missed labeling the fully connected layer (FC layer) in the final classification part of Figure 1. We will correct this in the final version of the paper. We appreciate your attention to detail.
>
> ### Limitations:
>
> 1. **The DARNet has only been validated under the subject-dependent condition, and it may not perform as well under subject-independent conditions.**
>
>    Recently, we conducted additional leave-one-subject-out cross-validation experiments on the publicly available DTU and KUL datasets, using a 1-second decision window that closely matches human attention shifts. The results showed that our model performed exceptionally well under subject-independent conditions, surpassing the performance of current SOTA models. The specific results are shown in Table 3:
>
>    Table3: Cross-subject experiment comparison on the KUL and DTU dataset for 1s. The GCN is currently the state-of-the-art (SOTA) model for cross-subject tasks.
>
>    | model            | KUL                               | DTU                               |
>    | ---------------- | --------------------------------- | --------------------------------- |
>    | SSF-CNN          | $54.1 \pm 6.60$                   | $48.7 \pm 3.96$                   |
>    | MBSSFCC          | $59.0 \pm 8.72$                   | $49.3 \pm 4.86$                   |
>    | DBPNet           | $59.7 \pm 8.12$                   | $53.7 \pm 5.98$                   |
>    | GCN[1]           | $64.4 \pm 6.36$                   | $53.5 \pm 7.53$                   |
>    | **DARNet(ours)** | $\textbf{74.0} \pm \textbf{11.4}$ | $\textbf{56.0} \pm \textbf{5.71}$ |
>
>    [1] S. Cai, R. Zhang and H. Li, "Robust Decoding of the Auditory Attention from EEG Recordings Through Graph Convolutional Networks," *ICASSP 2024 - 2024 IEEE International Conference on Acoustics, Speech and Signal Processing (ICASSP)*, Seoul, Korea, Republic of, 2024, pp. 2320-2324, doi: 10.1109/ICASSP48485.2024.10447633.

---

> > ### Comment · Reviewer_XzJY · 2024-08-08
> > **Thank you for your prompt and comprehensive response to my review.**
> >
> > Your detailed answers have effectively addressed my concerns. The additional cross-subject experiments are a valuable addition, significantly strengthening the overall contribution of the paper. Additionally, the authors have provided a thorough rationale, supported by ablation studies, demonstrating the effectiveness of their approach within the constraints of low-latency scenarios.
> > I am confident that this paper makes a significant contribution to the field of auditory attention detection and will be of great interest to the community. Therefore, I am raising my rating to "Strong Accept."

---

> > > ### Author Response · Authors · 2024-08-11
> > >
> > > We sincerely thank you for your positive feedback and for raising the rating of our paper. We greatly appreciate your recognition of our additional experiments and the overall contribution of our work. We are delighted that our paper is considered a significant contribution to the field and will be of interest to the community.

---

### Official Review · Reviewer_vVL1 · 2024-07-08

**Soundness:** 3
**Presentation:** 3
**Contribution:** 2
**Rating:** 6
**Confidence:** 3

**Summary:**

The paper proposes DARNet, a dual attention refinement network with spatiotemporal construction for auditory attention detection (AAD). The network captures spatiotemporal features and long-range latent dependencies from EEG signals, leading to improved classification accuracy and reduced parameter count compared to state-of-the-art models.

**Strengths:**

1. DARNet shows significant improvements in classification accuracy across multiple datasets, demonstrating its effectiveness in AAD tasks.
2. The model reduces the number of parameters by 14% compared to the state-of-the-art, which is beneficial for practical applications requiring efficient computation.
3. The integration of spatial and temporal convolutions to capture dynamic patterns in EEG signals enhances the model's ability to decode brain activity accurately.
4. The dual attention refinement module effectively captures long-range dependencies in EEG signals, addressing a common limitation in previous models.

**Weaknesses:**

1. The experiments are all subject-dependent, which may limit the generalizability of the results. Cross-subject conditions could be more relevant for practical applications, where the model needs to generalize across different individuals.
2. While the model achieves high performance, its complexity might make it difficult to interpret and understand the underlying mechanisms contributing to its success. More insights into the workings of the dual attention mechanism and its impact on EEG signal processing would be beneficial.

**Questions:**

NA

**Limitations:**

Limitations are discussed in the checklist.

---

> ### Author Rebuttal · Authors · 2024-08-06
>
> # Rebuttal:
>
> Thank you so much for your thoughtful comments and the time to provide constructive feedback!
>
> ## Weakness:
>
> 1. **Supplementary cross-subject experiment results:**
>
>    Recently, we conducted additional leave-one-subject-out cross-validation experiments on the publicly available DTU and KUL datasets, using a 1-second decision window that closely matches human attention shifts. The results showed that our model performed exceptionally well under subject-independent conditions, surpassing the performance of current SOTA models. The specific results are shown in Table 1:
>
>    Table1: Cross-subject experiment comparison on the KUL and DTU dataset for 1s. The GCN is currently the state-of-the-art (SOTA) model for cross-subject tasks.
>
>    | Model   | KUL                               | DTU                               |
>    | ------- | --------------------------------- | --------------------------------- |
>    | SSF-CNN | $54.1 \pm 6.60$                   | $48.7 \pm 3.96$                   |
>    | MBSSFCC | $59.0 \pm 8.72$                   | $49.3 \pm 4.86$                   |
>    | DBPNet  | $59.7 \pm 8.12$                   | $53.7 \pm 5.98$                   |
>    | GCN[1]  | $64.4 \pm 6.36$                   | $53.5 \pm 7.53$                   |
>    | DARNet  | $\textbf{74.0} \pm \textbf{11.4}$ | $\textbf{56.0} \pm \textbf{5.71}$ |
>
>    [1] S. Cai, R. Zhang and H. Li, "Robust Decoding of the Auditory Attention from EEG Recordings Through Graph Convolutional Networks," *ICASSP 2024 - 2024 IEEE International Conference on Acoustics, Speech and Signal Processing (ICASSP)*, Seoul, Korea, Republic of, 2024, pp. 2320-2324, doi: 10.1109/ICASSP48485.2024.10447633.
>
> 2. **More insights into the workings of the dual attention mechanism and its impact on EEG signal processing would be beneficial：**
>
>    We supplemented our study with ablation experiments using a single-layer attention refinement module, referred to as single-DARNet, while the original model is denoted as DARNet. The results, shown in Tables 1 and 2, reveal that the performance gap between single-DARNet and DARNet widens with increasing decision window lengths.
>
>    Our analysis suggests that with a short decision window length of 0.1 seconds, the EEG sequence contains few data points, making each point significantly impactful on the decoding accuracy. At this stage, the local features of the EEG signal play a more decisive role in the decoding process. Therefore, the performance difference between single-DARNet and DARNet is not substantial. However, under experimental conditions with medium to long decision window lengths, such as 1 second and 2 seconds, the performance gap between single-DARNet and DARNet gradually increases. This indicates that as the decision window lengthens, the long-range dependencies within the EEG signals increasingly influence the decoding accuracy. This experiment demonstrates that the long-range dependencies are captured within EEG signals. Additionally, under experimental conditions with even longer sliding window sizes, such as 5 seconds, our model continues to perform exceptionally well, further proving that the long-range dependencies are captured within EEG signals.
>
>    Table2: Ablation Study on DTU dataset.
>
>    | Model         | 0.1s                             | 1s                                | 2s                               | 5s                               |
>    | :------------ | :------------------------------- | :-------------------------------- | :------------------------------- | :------------------------------- |
>    | single-DARNet | $79.1 \pm 5.66$                  | $86.3 \pm 5.83$                   | $88.0 \pm 4.03$                  | $90.2 \pm 5.62$                  |
>    | **DARNet**    | $\textbf{79.5}\pm \textbf{5.84}$ | $\textbf{87.8} \pm \textbf{6.02}$ | $\textbf{89.9}\pm \textbf{5.03}$ | $\textbf{93.1}\pm \textbf{4.37}$ |
>
>    Table3: Ablation Study on KUL dataset
>
>    | Model         | 0.1s            | 1s              | 2s              | 5s              |
>    | :------------ | :-------------- | :-------------- | :-------------- | :-------------- |
>    | single-DARNet | $91.1 \pm 5.18$ | $95.5 \pm 3.28$ | $96.2 \pm 2.98$ | $96.7 \pm 4.03$ |
>    | **DARNet**    | $91.6\pm 4.83$  | $96.2 \pm 3.04$ | $97.2\pm 2.50$  | $98.0\pm 3.17$  |

---

> > ### Comment · Reviewer_vVL1 · 2024-08-09
> >
> > Thanks for your supplementary experiments and I have increased the score from 5 to 6.

---

> > > ### Author Response · Authors · 2024-08-11
> > >
> > > We sincerely appreciate your recognition of our supplementary experiments and your decision to increase the score.

---

### Official Review · Reviewer_FMXQ · 2024-07-08

**Soundness:** 3
**Presentation:** 3
**Contribution:** 3
**Rating:** 6
**Confidence:** 4

**Summary:**

This paper proposes a new architecture for auditory attention detection (AAD) that consists of three key components: 1) Convolutional layers applied to the temporal and spatial dimensions of EEG signals in a sequential manner to extract features. 2) Two attention layers to process these features. 3) A feature fusion module to combine the outputs of the two attention layers. The method was tested on three different datasets and achieved state-of-the-art (SOTA) classification results.

**Strengths:**

The paper presents an effective and innovative adoption of convolution and attention layers for auditory attention detection. The model has been rigorously validated on three different datasets. The inclusion of ablation studies further strengthens the validity of the findings by systematically analyzing the contribution of each component of the model.

**Weaknesses:**

* Using convolutional layers on the temporal dimension followed by the spatial dimensions of EEG signals is not a novel idea. Classic EEG+DL+BCI papers, such as ConvNet (Schirrmeister, 2017) and EEGNet (Lawhern, 2018), already employ similar concepts. Many EEG+DL+AAD papers also adopt spatial processing modules (e.g., references 3 and 14 in the paper). The author should provide a more in-depth discussion on why their spatial module is more effective than existing methods. Specifically, they should highlight any unique innovations or improvements their approach offers over previous works.
* It would be interesting to see the comparison between dual-attention layer and one attention layer.
* It is crucial to explicitly mention in the methods section that this is a subject-dependent study.
* There is a typo in the notations on row 91.

**Questions:**

How did 48 minutes of recording in the KUL dataset yield over 5000 1-second decision windows? If a sliding window with overlap was used, please describe the procedure in detail to ensure that there are no repeated signal segments in the training and test sets.

**Limitations:**

* Reproducibility of the paper would be significantly enhanced if the author provided the full code.

---

> ### Author Rebuttal · Authors · 2024-08-06
>
> # Rebuttal:
>
> Thank you so much for your thoughtful comments and the time to provide constructive feedback!
>
> ### Weakness:
>
> 1. **A more in-depth discussion on why their spatial module is more effective than existing methods.**
>
>    Auditory attention decoding requires processing EEG signals under complex acoustic stimuli. Temporal regularity is crucial in selective hearing, making the extraction of spatiotemporal relationships key to decoding. Traditional ConvNet models for motor imagery first perform temporal convolution and then spatial convolution, potentially overlooking complex interactions between spatial and temporal dimensions. Most current EEG+DL+AAD methods use DE features, such as transforming the temporal domain into the frequency domain and projecting it onto a 2D topological map, which can lose important temporal variations. Recent methods like STANet (Su et al.) explore spatial features in the temporal domain but are limited to assigning weights to EEG channels without considering nonlinear relationships between them.
>
>    + **Rich Spatiotemporal Feature Extraction:** We capture spatial dependencies between EEG channels and temporal patterns within each channel. By emphasizing channel interactions and using GELU to capture nonlinear relationships, we achieve comprehensive spatiotemporal representations.
>    + **Multi-level temporal pattern extraction:** We integrate a dual attention mechanism on the basis of effective convolution design, enabling the model to focus on distant time points and capture long-term attention pattern changes.
>
>    Additionally, we compared the performance of temporal-only, spatial-only, and spatiotemporal feature extraction. This comparison offers valuable insights for future research.
>
> 2. **The comparison between dual-attention layer and one attention layer.**
>
>    We have supplemented our study with ablation experiments using a single-layer attention refinement module, referred to as single-DARNet, while the original model is denoted as DARNet. The results, shown in Tables 1 and 2, reveal that the performance gap between single-DARNet and DARNet widens with increasing decision window lengths.
>
>    Our analysis indicates that with a short decision window of 0.1 seconds, the EEG sequence contains few data points, making each point significantly impactful on decoding accuracy. At this short window length, local EEG features are crucial, so the difference in performance between single-DARNet and DARNet is minimal. However, as the decision window lengthens to 1 and 2 seconds, the gap between single-DARNet and DARNet increases. This suggests that longer decision windows better capture the long-range dependencies within EEG signals, enhancing decoding accuracy.
>
>    Table1: Ablation Study on DTU dataset
>
>    | Model         | 0.1s            | 1s              | 2s              | 5s              |
>    | :------------ | :-------------- | :-------------- | :-------------- | :-------------- |
>    | single-DARNet | $79.1\pm 5.66$ | $86.3\pm 5.83$ | $88.0 \pm 4.03$ | $90.2 \pm 5.62$ |
>    | **DARNet**    | $79.5\pm 5.84$  | $87.8\pm 6.02$ | $89.9\pm 5.03$  | $93.1\pm 4.37$  |
>
>    Table2: Ablation Study on KUL dataset
>
>    | Model         | 0.1s            | 1s              | 2s              | 5s              |
>    | :------------ | :-------------- | :-------------- | :-------------- | :-------------- |
>    | single-DARNet | $91.1\pm 5.18$ | $95.5\pm 3.28$ | $96.2\pm 2.98$ | $96.7\pm 4.03$ |
>    | **DARNet**    | $91.6\pm 4.83$  | $96.2 \pm 3.04$ | $97.2\pm 2.50$  | $98.0\pm 3.17$  |
>
> 3. **Supplementary cross-subject experiment results:**
>
>    Recently, we conducted additional leave-one-subject-out cross-validation experiments on the publicly available DTU and KUL datasets, using a 1-second decision window that closely matches human attention shifts. The results showed that our model performed exceptionally well under subject-independent conditions, surpassing the performance of current SOTA model. The specific results are shown in Table 3.
>
>    Table3: Cross-subject experiment comparison for 1s. The GCN is currently the SOTA model for cross-subject tasks.
>
>    | Model      | KUL                               | DTU                               |
>    | :--------- | :-------------------------------- | :-------------------------------- |
>    | SSF-CNN    | $54.1 \pm 6.60$                   | $48.7 \pm 3.96$                   |
>    | MBSSFCC    | $59.0 \pm 8.72$                   | $49.3 \pm 4.86$                   |
>    | DBPNet     | $59.7 \pm 8.12$                   | $53.7 \pm 5.98$                   |
>    | GCN        | $64.4 \pm 6.36$                   | $53.5 \pm 7.53$                   |
>    | **DARNet** | $\textbf{74.0} \pm \textbf{11.4}$ | $\textbf{56.0} \pm \textbf{5.71}$ |
>
>
> 4. **There is a typo on row 91.**
>
>    We are very grateful to reviewer for reviewing the paper so carefully.
>
>    Modified version: By employing a moving window on the EEG data, we obtain a series of decision windows, each containing a small duration of EEG signals. Let $R=[r_1,...,r_i,...,r_N] \in \mathbb{R}^{T×N}$ represent the EEG signals of a decision window, where $r_i \in \mathbb{R}^{N \times 1}$ represents the EEG data at the $i$-th time point within a decision window, containing $N$ channels.
>
> ### Questions:
>
> 1. **48 minutes of recording yield over 5000 1s decision windows.**
>
>    The 48 minutes of recording contains 2880 seconds of EEG data. Consistent with previous studies, we set the repetition rate to 0.5. Thus, we obtain $2880 \times 2 = 5760$ 1s decision windows.
>
> 2. **Prevent data leakage.**
>
>    To prevent data leakage, we designated the first 90% of the EEG data from each trial as the training set and the remaining 10% as the test set. We then applied the sliding window technique separately to both sets.
>
> ### Limitations:
>
> 1. Reproducibility of the paper would be significantly enhanced if the author provided the full code.
>
>    We will upload the complete code once the paper is accepted.

---

> > ### Comment · Reviewer_FMXQ · 2024-08-07
> >
> > I thank the author for the detailed response. The cross-subject study improved the quality of the work. I changed my score to  6: Weak Accept

---

> > > ### Author Response · Authors · 2024-08-11
> > >
> > > We greatly appreciate your constructive comments and are pleased that our cross-subject study addressed your concerns.

---

### Official Review · Reviewer_eWL2 · 2024-07-31

**Soundness:** 3
**Presentation:** 3
**Contribution:** 2
**Rating:** 4
**Confidence:** 3

**Summary:**

The manuscript is aim to capture the and spatial distribution information and long-range dependencies in EEG signals. Two modules are designed to solve the upper two challenges, spatiotemporal construction module and dual attention refinement module. The experiment  have shown the superiority of the proposed method. And the analysis have shown the advantage of each module.

**Strengths:**

1)Good writing and organization.
2)Method is good understanding and can be replicated.
3)The experiment and analysis is sufficient.

**Weaknesses:**

1)The manuscript is lack of novelty.
2)It is less clarity in session Abstract on which dataset the accuracy is improved using different sliding window size.
3)There is no analysis that the level of computation cost and time consuming of the mentioned methods in Table 2.

**Questions:**

1)Many researchers have done this work, to capture the long-range dependencies and spatial distribution information within EEG signals. Is there any more innovation of your proposed method? (From framework design and experimental performance.)
2)What sliding window size is the best performance with? Have you ever try other size longer than 2 s?
3)Please prove that the long-range dependencies are captured within EEG signals.

**Limitations:**

Yes.

---

> ### Author Rebuttal · Authors · 2024-08-06
>
> # Rebuttal:
>
> Thank you so much for your thoughtful comments and the time to provide constructive feedback!
>
> ### Weakness:
>
> 1. **Innovation summary:**
>
>    We apologize for any confusion that may have led to the perception of a lack of novelty in our manuscript. We have clarified and summarized our key innovations as follows:
>
>    Firstly,our method highlights EEG channel interactions. Unlike some spatial feature explorations through the frequency domain that exist in the AAD field, our work effectively extracts spatial information directly from the temporal domain and uses GELU to capture nonlinear relationships. We compared temporal-only, spatial-only, and spatiotemporal feature extraction, offering significant insights for summarizing, inspiring, and guiding future works in AAD.
>
>    Secondly, unlike methods that stack multiple self-attention layers to capture long-range dependencies, we employ a refinement operation that progressively compresses the EEG signal sequence. This approach models global dependencies more effectively and reduces the model's parameter count. By halving the sequence length at each step, our method reduces the complexity of self-attention to one-fourth of the original.
>
>    Thirdly, our feature fusion module effectively balances local EEG features and long-range dependencies, enhancing decoding accuracy.
>
>    Finally, our model outperforms current SOTA models across multiple datasets while maintaining a lower parameter count.
>
> 2. **Dataset clarity in Abstract.**
>
>    Thank you for your thorough review. We realize we overlooked specifying the dataset annotations and will update this in the final version of the paper.
>
> 3. **The computation cost and time consuming of the mentioned methods .**
>
>    Table 1 includes data on computation cost and time consumption, conducted on the KUL dataset for 1s. The results demonstrate that our model has fewer parameters, as well as lower MACs and test time compared to current SOTA models.
>
>    Table1. DBPNet is currently the SOTA model.
>
>    | Model      | Param(M) | MACs(M)   | Test time(ms) |
>    | :--------- | :------- | :-------- | :------------ |
>    | MBSSFCC    | 83.91    | 89.15     | 11.20         |
>    | DBPNet     | 0.91     | 96.55     | 11.98         |
>    | **DARNet** | **0.78** | **16.36** | **10.03**     |
>
> ###  Questions:
>
> 1. **Is there any more innovation of your proposed method?**
>
>    See response to weakness1.
>
> 2. **What sliding window size is the best performance with? Have you ever try other size longer than 2s?**
>
>    Previous research shows that longer sliding windows capture more information and typically improve performance. However, in AAD tasks, longer windows increase latency, which is less suitable for human attention switching. Therefore, current research focuses on improving decoding accuracy with short decision windows, specifically 0.1s, 1s, and 2s. Our results for a 5-s window, presented in Table 2 & 3, confirm that our model continues to outperform current SOTA models.
>
>    Table2: AAD accuracy(%) comparison on DTU dataset.
>
>    | Model      | 0.1s             | 1s               | 2s              | 5s              |
>    | :--------- | :--------------- | :--------------- | :-------------- | :-------------- |
>    | MBSSFCC    | $66.9 \pm 5.00$  | $75.6 \pm 6.55$  | $78.7 \pm 6.75$ | $80.2\pm 8.64$  |
>    | DBPNet     | $75.1  \pm 4.87$ | $83.9  \pm 5.95$ | $86.5 \pm 5.34$ | $90.1 \pm 5.82$ |
>    | **DARNet** | $79.5\pm 5.84$   | $87.8\pm 6.02$   | $89.9 \pm 5.03$ | $92.4\pm 4.71$  |
>
>    Table3: AAD accuracy(%) comparison on KUL dataset.
>
>    | Model      | 0.1s            | 1s              | 2s              | 5s              |
>    | :--------- | :-------------- | :-------------- | :-------------- | :-------------- |
>    | MBSSF      | $79.0 \pm 7.34$ | $86.5 \pm 7.16$ | $89.5 \pm 6.74$ | $92.8 \pm 5.32$ |
>    | DBPNet     | $87.1 \pm 6.55$ | $95.0 \pm 4.16$ | $96.5 \pm 3.50$ | $97.0 \pm 4.05$ |
>    | **DARNet** | $91.6\pm 4.83$  | $96.2\pm 3.04$  | $97.2 \pm 2.50$ | $98.0\pm 3.17$  |
>
> 3. **Please prove that the long-range dependencies are captured within EEG signals.**
>
>    We supplemented our study with ablation experiments using a single-layer attention refinement module, referred to as single-DARNet, while the original model is denoted as DARNet. The results, shown in Tables 4 & 5, reveal the performance gap between single-DARNet and DARNet widens with increasing decision window lengths.
>
>    Our analysis indicates that with a short decision window of 0.1s, the EEG sequence contains few data points, making each point significantly impactful on decoding accuracy. At this short window length, local EEG features are crucial, so the difference in performance between single-DARNet and DARNet is minimal. However, as the decision window lengthens to 1s and 2s, the gap between single-DARNet and DARNet increases. This suggests that longer decision windows better capture the long-range dependencies within EEG signals, enhancing decoding accuracy. Our experiments with even longer sliding window sizes, such as 5s, confirm that our model effectively captures these long-range dependencies.
>
>    Table4: Ablation Study on DTU dataset.
>
>    | Model         | 0.1s            | 1s              | 2s              | 5s              |
>    | :------------ | :-------------- | :-------------- | :-------------- | :-------------- |
>    | single-DARNet | $79.1 \pm 5.66$ | $86.3 \pm 5.83$ | $88.0 \pm 4.03$ | $90.2 \pm 5.62$ |
>    | **DARNet**    | $79.5\pm 5.84$  | $87.8 \pm 6.02$ | $89.9\pm 5.03$  | $93.1\pm 4.37$  |
>
>    Table5: Ablation Study on KUL dataset.
>
>    | Model         | 0.1s            | 1s              | 2s              | 5s              |
>    | :------------ | :-------------- | :-------------- | :-------------- | :-------------- |
>    | single-DARNet | $91.1 \pm 5.18$ | $95.5 \pm 3.28$ | $96.2 \pm 2.98$ | $96.7 \pm 4.03$ |
>    | **DARNet**    | $91.6\pm 4.83$  | $96.2 \pm 3.04$ | $97.2\pm 2.50$  | $98.0\pm 3.17$  |

---

### Decision · Program_Chairs · 2024-09-25

**Decision:**

Accept (poster)

**Comment:**

Auditory and audio-visual attention tasks are quite difficult to model, characterize and enable prediction. Specifically, these are not well studied from a learnability point of view. The authors propose a model that combines deep signal processing / classical insights with established spatio-temporal modeling (convolutions in time with attention in space) to build a better than state of the art EEG-driven multi-modal model for attention. The resulting work also improves performance quite significantly compared to baselines / sota.

There is general agreements among reviewers regarding the impact, and I would suggest the authors to address / capture content from comments on rationale for proposed models' novelty, single vs. dual attention and a summary from additional experimental insights in the final version for publication.